# A Novel SNPs in Alpha-Lactalbumin Gene Effects on Lactation Traits in Chinese Holstein Dairy Cows

**DOI:** 10.3390/ani10010060

**Published:** 2019-12-29

**Authors:** Fan Yang, Manling Zhang, Yuewen Rong, Zaiqun Liu, Shuai Yang, Wei Zhang, Jun Li, Yafei Cai

**Affiliations:** 1College of Animal Science and Technology, Nanjing Agricultural University, Nanjing 210095, China; yangfanbridge@ahnu.edu.cn (F.Y.); 18895312558@163.com (M.Z.); weizhang@njau.edu.cn (W.Z.); 2College of Life Sciences, Anhui Provincial Key Lab of the Conservation and Exploitation of Biological Resources, Anhui Normal University, Wuhu 241000, China; liuzaiqun@126.com (Z.L.); yangshuai199410@163.com (S.Y.); 3Physical and chemical testing laboratory, Anji County Center for Food and Drug Control, Huzhou 313300, China; rongyuewen2006@163.com

**Keywords:** Alpha-lactalbumin (α-LA), PCR-SSCP, Chinese Holstein dairy cows, SNPs, lactose

## Abstract

**Simple Summary:**

Alpha-lactalbumin (α-LA) is a major whey protein component in mammalian milk, such as human (approximately 36%), bovine (approximately 17%), and other species, etc., It is involved in the regulation of lactose synthesis and has high nutritional value, especially in infant formula. Previous studies have confirmed that bovine α-LA gene 5′-flanking region has single nucleotide polymorphisms (SNPs), but little is known about polymorphisms in other regions, especially sequence coding for amino acids in protein (CDS) and their adjacent non-coding regions, including Chinese Holstein dairy cows. This study focused on investigated SNPs in the CDS and their adjacent non-coding regions of the α-LA gene in Chinese Holstein dairy cows, and assessed the association between SNPs and lactation traits. Sequence alignment showed that a potential SNPs (562th, G/A) in CDS2 region affect protein spatial structure, suggesting that this SNPs might affect the lactation traits of cows (milk type (Holstein and Jersey), and non-milk type (Bos Taurus)) need more in-depth study. More importantly, a novel SNPs at 1847th (T/C) bp in non-coding region near CDS4 was significantly associated with milk lactose composition, and lactose contents were significantly correlated with milk protein content, indicating that the SNPs could be used as a novel potential molecular marker for lactation traits in Chinese Holstein dairy cows.

**Abstract:**

Alpha-lactalbumin (α-LA) is a major whey protein in bovine and other mammalian milk, which regulates synthesis of lactose. Little is known about its genetic polymorphism and whether can be used as a potential marker for dairy ingredients, milk yield traits, and milk properties. To investigate its polymorphisms and their relationship with milk lactation traits in Chinese Holstein dairy cows, single-strand conformation polymorphism method (PCR-SSCP) and direct sequencing method were used to mark the α-LA gene SNPs. AA (0.7402) and AB (0.2598) genotypes were screened out by PCR-SSCP bands analysis in two independent populations. Direct sequencing revealed that there is one SNP at 1847th (T/C) bp in noncoding region of α-LA gene with highly polymorphic (0.5 < PIC = 0.5623 or 0.5822), of which T is in AA genotype while C in AB. Association analysis also showed that lactose content (*p* < 0.05) was negatively correlated with fat and protein contents within subgroup, indicating that the SNPs (1847th, T/C) in α-LA gene could be used as a novel potential molecular marker for lactation traits in Chinese Holstein dairy cows.

## 1. Introduction

Milk protein components have an important influence on milk production traits [1], and mainly affect by genetic factors [2,3]. There are six major milk protein components (α_S1_-CN, α_S2_-CN, β-CN, κ-CN, α-LA, and β-LG) in bovine milk [4,5]. However, α-LA, as an important whey protein, plays a vital role in lactose biosynthesis [6], which account for about 3.5% of total milk protein and 17% of whey protein [7], and in turn affects lactation yield [8]. However, α-LA protein is also considered as the most lactating-specific whey protein and the second highest protein in bovine whey protein concentrate, and whey protein isolates as well. Further, α-LA products have similar physical properties to whey protein, including high-quality protein, high-water solubility, good thermal stability, etc., which can be used in food processing industry. As a nutritional supplement, α-LA has attracted more and more attention in its potential application in human health [7]. It is also added as a supplemental protein in infant formula to improve intestinal health, enhance immunity, promote growth and facilitate the absorption of needed trace elements such as iron and zinc, and contributes to infant development [7,9]. In addition, due to its unique tryptophan content, α-LA also has the potential to be a nutritional supplement that supports adult nerve function and sleep [7], and has further potential applications in cancer treatment or enhanced immune response [10,11,12]. Therefore, α-LA protein has high potential application value for human nutrition and health.

The α-LA gene genetic polymorphisms have been reported in many species, including goat [13], pig [14], buffalo [15], Holstein and Nellore cows [16]. Dettori et al. used (single-strand conformation polymorphism) SSCP and sequencing studies to detect the single nucleotide polymorphism in the 5′ flanking region and 3′ UTR of the Sarda goats α-lactalbumin precursor gene (LALBA) associated with milk production, components, and renneting properties [17]. In addition, previous studies have also shown that the 5′-flanking region polymorphisms of α-LA protein gene are associated with the difference of α-LA gene expression, which might be related to milk yield and composition in Holstein cows [8,18], such as the allelic frequencies of α-LA locus -1689th and +15th polymorphism in Holstein and Nellore [16]. Moreover, buffalo α-LA gene exons 1/2 and Holstein-Friesian cows α-LA gene exon 2 have SNPs leading to amino acid structural changes [15,19], suggesting that the genetic polymorphisms of α-LA gene affect the lactation performance traits of dairy cows. However, there are few reports on the polymorphisms of the exon and intron regions in α-LA gene of the Chinese Holstein cows, so the association between milk traits (milk yield and components) and SNPs remains unclear. Therefore, the polymorphism of α-LA gene CDSs and their adjacent regions provide an alternative breeding opportunity for improving milk protein traits in Chinese Holstein dairy cows. Importantly, it also provides reliable genetic parameter estimates for accurate assessment of breeding potential.

In this study, four pairs of primers for PCR amplification CDSs regions (including parts noncoding regions) of the α-LA gene were detected via SSCP typing in Chinese Holstein dairy cows, and then SNPs were identified by directly sequencing PCR products. Next, the association between the above SNPs of the α-LA gene and milk composition and yield was analyzed. The aim of this study attempts to provide more molecular genetic information for finding markers that are closely related to the lactation traits of Chinese Holstein dairy cows.

## 2. Materials and Methods

### 2.1. Animal and Preparation

The experimental animals and their operating procedures in this study were in compliance with the regulations of national and local animal welfare agencies, and were approved by the Animal Protection and Welfare Committee of the Cow Research Center of Nanjing Agricultural University (Approval No.20171229).

A total of 535 Chinese Holstein dairy cows from two pastures of the same Dairy Company (Nanjing Weigang Dairy Co., Ltd.) were used as experimental cows in this study, of which 152 dairy cows were screened from one pasture, while 383 concurrent cows were selected in another pasture. All animals have same parity and lactation, similar growth environments and equivalent nutritional conditions. To minimize damage to the cows, blood samples were collected from bovine tail vein using a 5 mL EDTA-K2 sterile blood collection device. Genomic DNA extracted from blood using genomic DNA extraction kit (#B518223, Sangon Biotech, Shanghai, China). The quality (OD260/OD280 = 1.7–1.8) of bovine whole blood genomic DNA were detected by the SparkTM multimode microplate reader (Tecan, Salzburg, Austria). Track and record the dairy herd improvement (DHI) data and the key milk indices for association analysis. DHI test: From the fifth day after the cow delivery, to the end of this lactation period. The general test interval is 21–42 days, and a lactation period requires 9–10 tests. To facilitate statistics and management, DHI test selects 30-day centralized collection of milk-like samples.

### 2.2. PCR-SSCP Analysis

Based on the DNA sequence (NCBI Accession Number: AF249896), we successfully designed four pairs of primers for PCR amplification of the α-LA gene CDS regions (Table 1). The final volume of the PCR reaction was 12.5 μL, including 1.25 μL 10 × buffer (with Mg^2+^), 0.25 μL dNTPs, 0.35 μL upstream and 0.3 μL downstream primer, 0.1 μL Taq DNA polymerase, 50 ng DNA template and 8.75 μL ddH_2_O. PCR reaction conditions were as follows: denaturation 94 °C for 4 min, 35 cycles of 94 °C for 40 s, 58.2 °C for 30 s and 72 °C for 50 s, and extension at 72 °C for 7 min. PCR products detected using 2% agarose gel electrophoresis. Then SSCP typing were performed via non-denatured polyacrylamide gel electrophoresis with single strand DNA (5 μL of the PCR products mixed with the denaturant (1:1) and separated on a 12% polyacrylamide gel, then stained with AgNO_3_, finally the genotype and allele frequencies were directly counted).

### 2.3. PCR Products Sequencing and Alignment

PCR products with different SSCP bands were purified and single nucleotide polymorphisms (SNPs) of different SSCP bands were detected by direct sequencing. Sequencing results were aligned in the NCBI library to determine accurate amplification of the CDSs and their adjacent regions in the α-LA gene of dairy cows. DNA*star* 7.0 software (DNASTAR, Inc., Wisconsin, USA) was used to analyze the diversity and homology of different genotype sequences, and the nucleotide substitutions compared and analyzed as well.

### 2.4. Bovine α-LA Protein Spatial Structure Prediction Modeling

Template (amino acid sequence) search with BLAST and HHBlits had been performed against the SWISS-MODEL template library. Template search: the templates with the highest quality were selected for model construction based on the feature prediction templates aligned by the target templates. Model building: build models aligned with the target templates using ProMod3. Coordinates between the target and the templates copied from the templates to the models. Insert and delete refactored using the fragment library. Then, rebuild the side chain. Finally, the force field was used to regularize the geometry of the models. Model quality estimation: the overall models and the quality of each residual model was evaluated using the QNEAN scoring function, and then PDB file visualized in Cn3D software [20,21,22,23].

### 2.5. SNPs Association Analysis

We assumed that there is a true SNP locus associated with milk traits for each genotype of the CDSs regions (including parts noncoding regions) in α-LA gene. If the region contains SNPs, then the most significant related SNPs are most likely causal SNPs. Linear models are a common method for the correlation analysis of phenotypes and genotypes. Strict quality control was used to remove poorly performing SNP marker loci in SSCP typing. First, the following GLM procedure equation based on phenotype was constructed:
*y_ikj_* = *μ* + T*_i_* + C*_k_* + G*_j_* + *e_ikj_*(1)
where *y_ikj_* is a vector of phenotype for individual; *μ* is a vector of the overall mean of traits phenotypes; T*_i_* is a vector effect of the *i*th birth; C*_k_* is a vector of the *k*th environment; G*_j_* is the vector effect of the *j*th genotype at α-LA gene locus, and *e_ikj_* is random residual effect.

## 3. Results

### 3.1. The PCR Amplification Products of the Primer 4 have SSCP Polymorphism

The results of PCR amplification products showed that four CDS and their adjacent noncoding regions of α-LA gene were successfully obtained in Chinese Holstein dairy cows (Figure 1A). However, it’s noticed that three primers (primer 1 to primer 3) amplified fragments did not show differential SSCP typing bands (Figure 1B–D), suggesting that there are no SNPs in their PCR amplification products. There were only two different SSCP bands in amplified fragments of primer 4, and which were named AA and AB genotype, respectively. Although no BB-type bands were detected in the two independent populations, the above SSCP typing results were sufficient to indicate that there were polymorphisms in the primers four amplified fragments of the α-LA gene (Figure 1E). Therefore, the genotyping and allele frequencies of α-LA were counted based on the difference in SSCP bands of the primer 4 PCR products (Table 2). The AA and AB genotype frequencies in group 1 (*p* < 0.01) were 0.7566 and 0.2434, respectively, while in group 2 they were 0.7337 and 0.2663. Data also showed that allele A had genetic advantage in two independent Chinese Holstein populations (average value: A = 0.8701 > B = 0.1299).

### 3.2. Alignment Analysis of the Four Primers’ PCR Amplification Products

Three Bovine α-LA protein gene [NCBI Sequence ID: AB052166 (Holstein), AB052163 (Jersey), AF249896 (Bos Taurus)] were selected in the NCBI library for sequence alignment. Primer 1 PCR products sequencing results suggested that there is a potential SNPs in CDS 1 region (126th, G/A; the base number sites in the manuscript were based on the Holstein Sequence ID; similarly hereinafter) of the α-LA gene in Bos Taurus (Appendix A). At this locus, Chinese Holstein has the same base as Jersey and Holstein (base G), but unlike Bos Taurus (base A), there is a base transversion. Primer 2 product sequencing showed that there have a base deletion at 527th locus (A/-) in intron region of LA gene in Chinese Holstein (Appendix A). In this region, Bos Taurus has five loci (1190th (Bos Taurus)/535th (Holstein), 1193th/538th, 1217th/562th, 1389th/734th and 1406th/751th) different from Holstein, Jersey and Chinese Holstein. In addition, the locus (1217th/562th) appeared base transversion (A/G) in CDS2 region, and the change of nucleotide did not affect the amino acid alteration. Primer 3 PCR products near the CDS3 region of the intron, Bos Taurus has five consecutive bases (A/T, G/A, A/T, G/A, and G/A (1823th/1168th–1828th/1173th)) which are different from the other three bovine (Appendix A).

SNPs (1847th, T/C) were detected in primer 4 PCR products of α–LA gene noncoding region in Chinese Holstein (Figure 2). It’s also noted that CDS4 regions α-LA gene of the Chinese Holstein has the same sequence as Jersey, Holstein and Bos Taurus. It’s observed that there are two difference loci (1833th/1832th, T/G; 1857th/1858th, G/T) in noncoding region near CDS4 of the α–LA gene between these three cows (Holstein, Jersey and Bos Taurus) and Chinese Holstein dairy cows (Appendix A).

### 3.3. SNPs of CDS 2 Region (562th, G/A) in the α-LA Gene Affect the Spatial Structure

Two potential SNPs (126th, G/A; 562th, A/G) were in bovine α–LA gene. The potential SNPs appeared in CD 1 region did not affect amino acid residue change. However, the potential SNPs appeared in CDS 2 region (562th, A/G) of the α-LA gene, and this base transversion resulted in amino acid residue alteration (Ala/Thr) (Figure 3), which influenced the change of spatial structure of bovine α-LA (Figure 4). At the secondary structure level, α-helix appeared in the 72th, 73th amino acid residues in the Bos Taurus α-LA, and no α-helix found in the Holstein and other dairy cows α-LA. The tertiary spatial structure of the bovine (between Holstein and Bos Taurus) α-LA also changed significantly. Z-score, Local Quality Estimate (Chain A) and QMEAN6 Score used for quality assessment of α-LA spatial structure modeling (Appendix A).

### 3.4. Population Genetic Analysis Parameters of α-LA Gene SNPs (1847th, T/C) in Chinese Holstein Dairy Cows

SSCP and direct PCR sequencing showed that the α-LA gene of Chinese Holstein cows have SNP at 1847th. The population genetic analysis parameters (genetic variation statistics, gene diversity and polymorphism information content) of two independent subgroups of LA protein genes in Chinese Holstein cows were counted (Table 3). The SNPs (1847th) belonged to base transversion (T/C), effective allele number (1.600 and 1.648) and genetic diversity convergence (0.3750 and 0.3932) (*p* < 0.05). Polymorphism information content (PIC) was 0.5623 and 0.5822 in both subgroups, respectively. F-statistics values (natural selection statistics) were 0.6250 and 0.6098. And their 95% confidence intervals corresponding to them were (0.5022–0.9869) and (0.5052–0.9948). The polymorphism of α-LA protein gene affected the correlation between important milk components in both groups of Chinese Holstein cows (Table 4). Correlation analysis showed that there was a moderate intensity correlation (0.4 < total Pearson correlation value (PCV = 0.49) < 0.6) between the genotype and milk lactose components, but no correlation or weak correlation with other milk components (Fat content and Protein content) and milk 305D yield. Data also showed that lactose was negatively correlated with fat content (*p* < 0.01, PCV = −0.136, extremely weak correlation) and protein content (*p* < 0.01, PCV = −0.26, weak correlation), and no correlated with milk 305D yield (*p* = 0.309 > 0.05). Interestingly, when the nucleotide base was ‘T’, lactose negatively correlated with milk protein (*p* < 0.01, PCV = −0.226, weak correlation) and fat rates (*p* < 0.01, PCV = −0.156, extremely weakly correlation), and protein rates positively correlated with milk 305D yield, but *p* = 0.19 > 0.05 (no correlation), While was ‘C’, lactose only negatively correlated with milk protein (*p* < 0.01, PCV = −0.36, weak correlation), and no correlated with fat content and milk 305 yield. Chi-square test results indicated that milk fat, milk protein and 305D yield (*p* > 0.05) were all normally distributed, while lactose content (*p* < 0.05) was non-normally distributed.

## 4. Discussion

Polymorphism and milk quality studies have been widely used in the optimization of dairy population, and the link between milk composition and genetic variation in milk production has a profound impact on dairy production [2,3,24,25]. Recently, Poulsen et al. found that α-LA is a major component, which has negative effect on milk coagulation in Danish Jersey [26]. GWAS data also showed that SLC37A1 gene might be associated with α-LA phenotypes in multiple breed, and might affect milk yield [4]. Hereby, we highlighted the effects of SNPs in CDSs and their adjacent noncoding regions in α-LA gene on lactation traits of Chinese Holstein, especially lactose content and milk yield.

### 4.1. SSCP Typing Analysis Identified Two Genotypes (AA and AB Genotype) of α-LA Gene in Chinese Holstein Dairy Cows

PCR-SSCP typing bands showed that only the primer 4 amplified fragments had two differential bands in α-LA gene, and named AA and AB genotype, indicating that there are genetic polymorphisms in CDS4 and its adjacent regions of α-LA gene in Chinese Holstein dairy cows. Interestingly, only the above two genotypes (without BB genotype) were detected in the two subgroups of Chinese Holstein cows, suggesting that there are two alleles (A and B) in α-LA gene of Chinese Holstein. However, Dayal et al. [15] detected the presence of four alleles (A, B, C, and D) in riverine buffalos α-LA gene via SSCP method, while Martins et al. also confirmed the presence of alleles in the Holstein and Nellore α-LA protein genes by used RFLP method [16], showing that the number of α-LA gene alleles in Chinese Holstein are lower than the other three cows, even lower than Chinese dairy goats [α-LA gene (CC, CT and TT)] [13], indicating that the artificial breeding degree of Chinese Holstein cows may be higher, resulting in low polymorphism.

### 4.2. Genetic Polymorphisms (SNPs) of the α-LA Gene Existing in Four Different Bovine Breeds

There are significant differences in the distribution of α-LA gene polymorphisms in different bovine breeds [16,27,28]. Our direct sequencing results of Chinese Holstein α-LA gene were aligned with the sequences of three other kinds of cows [Sequence ID: AB052166 (Holstein), AB052163 (Jersey), AF249896 (Bos Taurus)] in NCBI database, indicating that the α-LA gene were significantly different between bovine breeds. The alignment with the Holstein sequence showed that the Chinese Holstein had a base (A>/) deletion at 527th locus, and detected a potential polymorphic site (G/T) at 1832th bp. Unlike the finding of Visker et al. [19] (one SNPs in exon 2 of the α-LA gene and causes amino acid change in Holstein-Friesian cows), we were concerned that the CDSs regions of the α-LA gene between Holstein and Jersey cattle were identical: shared two polymorphic sites [(126th, A/G) and 562th, G/A)] with Bos Taurus; one of them (562th, G/A)) caused amino acid changes, and ultimately led to changes the α-LA protein spatial structure. Whether this change in spatial structure affected α-LA protein gene function and milk production traits in Chinese Holstein required further verification. In addition, two potential SNPs (1832th (T/G) and 1858th (G/T)) were also detected in the intron of α-LA gene between Holstein and Jersey cows. Considering that both cows α-LA gene CDS regions are the same, it’s also necessary to verify whether the above two polymorphisms in noncoding regions associated with milk components, especially milk fat contents and yield.

### 4.3. One SNPs (1847th (T/C)) Locus in the Noncoding Region Near CDS4 of α-LA Gene Affected the Milk Lactose Content and Yield in Chinese Holstein Dairy Cows

Multiple SNPs have been confirmed in the α-LA genes of domestic animals, such as cattle and goat [13,16,27], riverine buffalo (SNPs in CDS 1 and 2 regions of α-LA gene) [15]. However, all four primers PCR products direct sequencing further confirmed that there were no SNPs in the CDS regions of α-LA gene in Chinese Holstein cows, but a novel SNPs were detected in the noncoding region (1847th, T/C) near the CDS4. Interestingly, we only detected T or C in the two subgroups of Chinese Holstein and they were genetically highly polymorphic (0.5 < PIC = 0.5623 and 0.5822), not found T/C heterozygous type. The association analysis showed that this polymorphism had a significant correlation with the lactose content of milk components in the two subgroups (*p* < 0.01, PCV = 0.49, moderate intensity). We also noted that lactose content significantly (*p* < 0.01) associated with milk fat and milk protein rate, suggesting that the genetic polymorphism of α-LA gene (1847th, T/C) might affect the milk composition in Chinese Holstein dairy cows.

## 5. Conclusions

The aim of this study was to investigate the α-LA gene SNP, and explore α-LA gene polymorphism association with milk composition and yield in Chinese Holstein dairy cows. Direct sequencing and sequence alignment showed that there are potential polymorphisms at 126th (A/G) in CDS1 and 562th (G/A) in CDS2, and the 562th polymorphism affects the change of protein spatial structure, suggesting that this SNPs might affect the lactation traits of cows (milk type (Holstein, Jersey, etc.,) and non-milk type). PCR-SSCP study and direct sequencing revealed that a base deletion at 527th (A/-) bp near CDS2 and one SNPs at 1847th (T/C) bp in noncoding region near CDS4 of α-LA gene in Chinese Holstein, and association analysis showed that there were significant correlation (*p* < 0.01, moderate intensity) between the SNPs (1847th) and lactose composition. Data also showed that lactose contents were significantly correlated with milk protein content, indicating that the SNPs (1847th, T/C) of α-LA gene could be used as a novel potential molecular marker for lactation traits in Chinese Holstein dairy cows.

## Figures and Tables

**Figure 1 animals-10-00060-f001:**
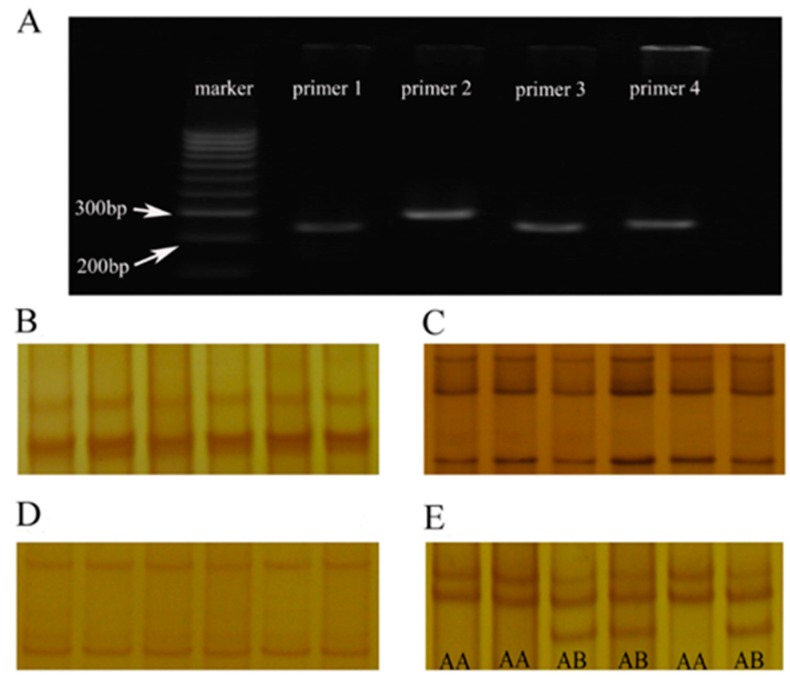
Amplified fragment and SSCP analysis of α-LA gene CDS and their adjacent regions in Chinese Holstein dairy cows. (**A**) 2% agarose gel electrophoresis of PCR amplification products (four pairs of primers) for CDS and their adjacent regions of α-LA gene. (**B**–**E**) SSCP analysis of PCR products for CDS and their adjacent regions of α-LA gene: (1) the primer 1, primer 2, and primer 3 PCR products had no difference in the SSCP analysis bands; (2) the primer 4 PCR products had two different bands, named AA and AB.

**Figure 2 animals-10-00060-f002:**
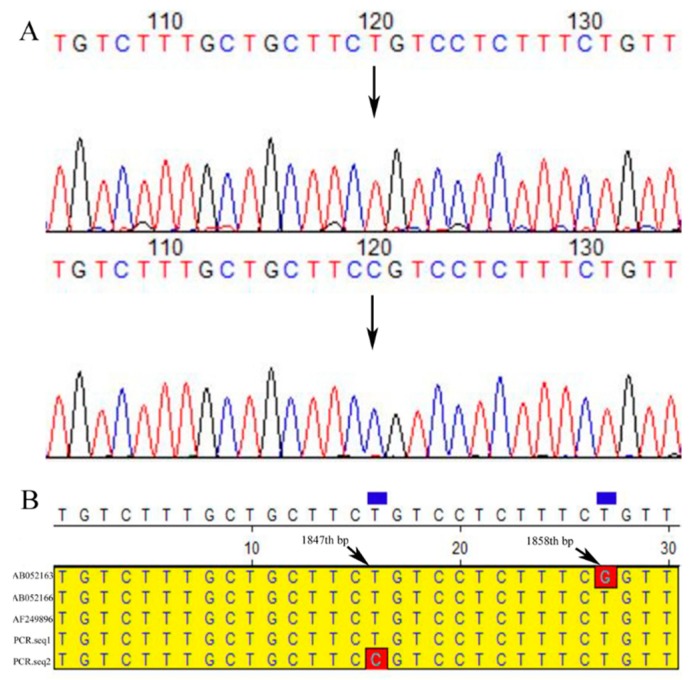
The 4th pair of primer products were directly sequenced and aligned in the NCBI database. (**A**) The α-LA gene polymorphism locus (black arrow mark) and its nearby sequences in Chinese Holstein cows. (**B**) The sequence near the LA gene polymorphism site of Chinese Holstein cows aligned with the three bovine sequences (AF249896 (Bos Taurus), AB052166 (Holstein) and AB052163 (Jersey)) in the NCBI library. One locus (red frame marker, 1847th) was the polymorphism of α-LA gene in Chinese Holstein cows. Another site (red frame marker, 1858th) was the differential locus of α-LA gene between Jersey and Holstein.

**Figure 3 animals-10-00060-f003:**
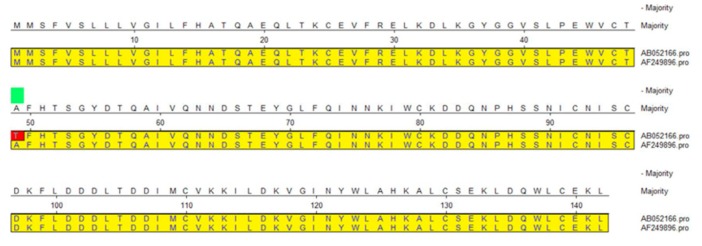
Bovine α-LA amino acid residue sequence alignment between Holstein (AB052166) and Bos Taurus (AF249896). Potential SNPs (562th, A/G) in the CDS2 region affects amino acid sequence changes (Ala/Thr) between Holstein and Bos Taurus (red marker).

**Figure 4 animals-10-00060-f004:**
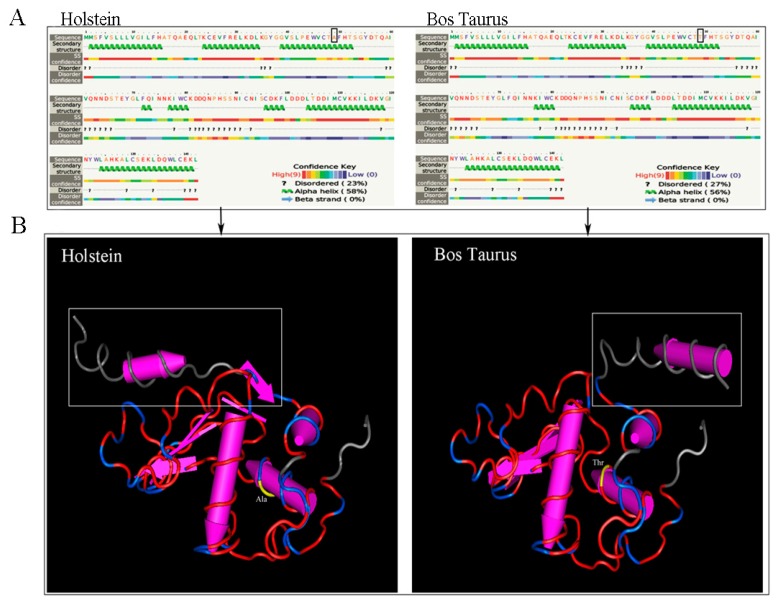
Bovine α-LA spatial structure (secondary and tertiary structure) analysis and alignment between Holstein and Bos Taurus. (**A**) Secondary structure of bovine α-LA between Holstein (left) and Bos Taurus (right) were changed, including α-helices and β corners; (**B**) Tertiary spatial structure of the α-LA changes significantly (box mark) between Holstein (left) and Bos Taurus (right).

**Table 1 animals-10-00060-t001:** Conventional PCR amplification primers for the four CDS and their adjacent regions of the α-LA gene (AF249896).

Region	Primer Name	Sequence (5′–3′)	Base Number (bp)	CDS Region (bp)	Annealing Temperature (°C)
CDS1	primer1F primer1R	GAGCAGTGTGGTGACCCC TGGAGGGAAAGAGTGAAGAG	237 (706–942)	751–883	58.7
CDS2	primer2F primer2R	GGTCTGGGAATACAGGTCC GGTTATCCCAGGAGTAGGTT	295 (1129–1423)	1205–1363	58.9
CDS3	primer3F primer3R	GGCAACAGGCATAAGCCTC GGACTGAGAAGAAAGAGAGG	240 (1727–1965)	1837–1912	58.7
CDS4	primer4F primer4R	CCTCAGCCTTCCTGGGGA CAGGGCTCAGAGACGAGTT	255 (2332–2586)	2417–2477	58.8

**Table 2 animals-10-00060-t002:** SSCP studied the genotype and allele frequency of the α-LA gene in two subgroups of Chinese Holstein cows.

Items		Genotypic Frequencies		Allele Frequencies		*p* < 0.01
		AA	AB	A	B	
Group 1	Number	115	37			
	Rates	0.7566	0.2434	0.8783	0.1217	
Group 2	Number	281	102			
	Rates	0.7337	0.2663	0.8468	0.1332	
Total	Number	396	139			
	Rates	0.7402	0.2598	0.8701	0.1299	

**Table 3 animals-10-00060-t003:** Population Genetic Analysis of α-LA Protein Genes (1847th) in Chinese Holstein Dairy Cows.

Groups	Sample Size	Na	Ne	H	PIC	Obs.F	Min F	Max F	L95	U95
Group 1	152	2	1.600	0.3750	0.5623	0.6250	0.50	0.9869	0.5022	0.9869
Group 2	383	2	1.648	0.3932	0.5822	0.6098	0.50	0.9948	0.5052	0.9948
Mean		2	1.624	0.3841	0.5723	0.6174	0.50	0.9908	0.5037	0.9908
SD		0	0.024	0.0091	0.00995	0.0076	0	0.00395	0.0015	0.00395

Note: Na: Observed number of alleles; Ne: Effective number of alleles; H: Gene diversity; PIC: polymorphism information content; Obs.F: F-statistics; L95-U95: 95% confidence interval of F-statistics.

**Table 4 animals-10-00060-t004:** Statistical analysis the milk components (mean fat and protein contents, Lactose) and 305D yield of different genotypes in Chinese Holstein dairy cows.

Items	Genotype/Base	Mean Fat Content (%)	Mean Protein Content (%)	Lactose (%)	305D (kg)
Group 1	AA/T	3.545 ± 0.3215	3.280 ± 0.3050	4.668 ± 0.232 **	10058.68 ± 822.96
	AB/C	3.602 ± 0.3426	3.125 ± 0.3062	4.732 ± 0.233 *	9723.01 ± 1033.12
Group 2	AA/T	3.574 ± 0.3968	3.155 ± 0.3040	4.793 ± 0.1821 **	10187.31 ± 940.89
	AB/C	3.520 ± 0.4747	3.200 ± 0.3008	4.837 ± 0.2077 **	10051.50 ± 1135.12
Mean ± SD		3.560 ± 0.3839	3.191 ± 0.3058	4.757 ± 0.2139	10005.27 ± 983.03

Note: * Correlation is significant at the 0.05 level; ** Correlation is significant at the 0.01 level.

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
