# Peer review of "A Novel SNPs in Alpha-Lactalbumin Gene Effects on Lactation Traits in Chinese Holstein Dairy Cows"

_animals, 2019, doi:10.3390/ani10010060_

Round 1
Reviewer 1 Report
The study is quite original but results and discussion should be better presented.
Specific concerns:
L19: check spaces
L24: association -> associated
L60: “In 2015, Dettori and his colleagues…” replace with Dettori et al.
L92: Delete “And”
L100-105: check spaces and units (i.e. 12.5μl -> 12.5 µL)
L156: The Figure legend is not clear, could you please clarify?
L166: Check the punctuation
L167: Which is the Figure S1???
L170: Which is the Figure S2???
L175: Which is the Figure S2
L177: Which is the Figure S4???
L188: Delete ‘’And’’
L191-193: There is not correspondence between CDS region reported in the text and the CDS region reported in the Figure legend, please check.
L195-196. ‘’Changes in amino acid residues (Ala/Thr) affected the α-LA protein spatial structure’’… It is redundant, please delate.
L197: Which is the Figure S5???
L199: The Figure legend is not clear, could you please clarify?
L231: “Poulsen et al and his colleagues …” replace with Poulsen et al.
L244: “Dayal and his colleagues” replace with Dayal and at.
L258: “Visker and his colleagues” replace with Visker et al.
Author Response
Dear Dr Reviewer:
First of all, thank you very much for your professional and objective comments. Those comments are all valuable and very helpful for revising and improving our paper, as well as the important guiding significance to our researches. We have revised our manuscripts in accordance with your suggestions and have made correction which we hope meet with approval, including grammatical errors, false tenses, and low-quality sentences, and revised the results, discussion and conclusion, etc. However, these changes will not influence the content and framework of the paper. Revised portion are marked in red in our manuscript. Please see the attachment.
We appreciate for your warm work earnestly, and hope that the correction will meet with approval.
Once again, thank you very much for your comments and suggestion.
Best regards and Happy New Year.
Yours sincerely
Fan Yang

Reviewer 2 Report
For keywords, include; Chinese Holstein Dairy cows; and thought text, include cows, or dairy cows, instead of just Holstein or Jersey.
Regarding dairy animals used, the age of cows and dietary ingredients have a dramatic influence on milk yield and milk components, in addition to pastures were any standard additional supplements used, like grains, minerals, and vitamins?
Also, time of year, like spring or winter will influence milk yield when the blood samples were obtained. Regarding the time of blood sampling, was the time standardized? ----stage of gestation, sampling before, after, or during milking?
Material and Methods regarding laboratory analysis:
PCR-SSCP analysis, PCR product sequencing, and alignment, Bovine a-LA protein spatial structure prediction, SNPs association analysis were adequately described and carried out.
Regarding ---3.3. SNPs of CDS 1 region (126th, G/A) in the a-LA gene affect the spatial structure on page 6, Figure 3 is difficult to read, may want to increase the print and decrease the size of the structures at bottom of figure 3.
Regarding a-LA gene genetic polymorphisms that have been studied in dairy cows (females), how would the results differ in males (Bulls) that sired the dairy cows used in this study? Just a curious question as any change in the use of genetics is most dramatically influenced by males (Bulls), as their genetics would also have an influence on milk production. May be an interesting study in the future?
Author Response
Dear Dr Reviewer 2:
First of all, thank you very much for your professional and objective comments. We fully appreciate and agree with your professional and objective expression. Your comments are all valuable and very helpful for revising and improving our paper, as well as the important guiding significance to our researches. We have revised our manuscripts in accordance with your suggestions and have made correction which we hope meet with approval, including grammatical errors, false tenses, and low-quality sentences, and revised the results, discussion and conclusion, etc. However, these changes will not influence the content and framework of the paper. Revised portion are marked in red in our manuscript. Please see the attachment.
We appreciate for your warm work earnestly, and hope that the correction will meet with approval.
Once again, thank you very much for your comments and suggestion.
Best regards and Happy New Year.
Yours sincerely
Fan Yang

Reviewer 3 Report
Materials and methods section:
Please provide more information about data collection on milk production traits.
Table 1: set a new column to indicate the start and the end of each CDS region.
Results:
L142: use Figure 1B-1D instead of 1B-1E.
Concerning results on genotype - milk production:
Table 4: set a new column for the number of cows for each genotype group
Table 4: authors found correlations but it is not clear that between which milk production traits (one of them is lactose). They say that the correlation is significant, but what is the strength of correlation?
The effect of genotype on milk production should be calculated. Authors also mention: "(1847th (T/C)) might affect the milk composition and yield"...You have genotype and milk comp. and yield data, thus it should be statistically confirmed.
L289-290: Did you found correlation between SNPs and lactose composition in milk?
Author Response
Dear Dr Reviewer 3:
First of all, thank you very much for your professional and objective comments. We fully appreciate and agree with your professional and objective expression. Your comments are all valuable and very helpful for revising and improving our paper, as well as the important guiding significance to our researches. We have supplemented the results of correlation analysis (between SNPs and milk production traits) in our manuscripts. We also have revised our manuscripts in accordance with your suggestions and have made correction which we hope meet with approval, including grammatical errors, false tenses, and low-quality sentences, and revised the results, discussion and conclusion, etc. However, these changes will not influence the content and framework of the paper. Revised portion are marked in red in our manuscript. Please see the attachment.
We appreciate for your warm work earnestly, and hope that the correction will meet with approval.
Once again, thank you very much for your comments and suggestion.
Best regards and Happy New Year.
Yours sincerely
Fan Yang

Round 2
Reviewer 1 Report
Authors opportunely replied to all requests
Manuscript is now acceptable for publication
Reviewer 3 Report
The paper was significantly improved and I accept this version of manuscript for publication.